# (Re)Construction of Quantum Space-Time: Transcribing Hilbert into Configuration Space

**DOI:** 10.3390/e26030267

**Published:** 2024-03-18

**Authors:** Karl Svozil

**Affiliations:** Institute for Theoretical Physics, TU Wien, Wiedner Hauptstrasse 8-10/136, 1040 Vienna, Austria; karl.svozil@tuwien.ac.at

**Keywords:** space-time frames, synchronization, induced relativity, quantum space-time

## Abstract

Space-time in quantum mechanics is about bridging Hilbert and configuration space. Thereby, an entirely new perspective is obtained by replacing the Newtonian space-time theater with the image of a presumably high-dimensional Hilbert space, through which space-time becomes an epiphenomenon construed by internal observers.

## 1. It-from-Click Imaging

This paper continues efforts to address the implications of quantum entanglement in the absence of gravitation for the construction of space-time coordinate frames. Previous papers have focused on context communication costs for simulating uniform quantum correlations [1] and conducted a detailed analysis of the violation of Boole’s conditions of possible (classical) experience by quantum mechanics [2].

Physical categories and conceptualizations, such as time and space, are formed in minds in accordance with the operational means available to observers. They are, thus, idealistic [3] and epistemic and, therefore, historic, preliminary, contextual, and not absolute.

Operationalists such as Bridgman [4], Zeilinger [5,6], or Summhammer [7] have emphasized the empirical aspect of physical category formation [8]. Hertz also highlighted the idealistic nature of physical ‘images’ (or mental categories) that internal observers construct to represent observations, and how these formal structures should remain consistent with, and connected to, empirical events or outcomes [9]: “We form for ourselves images or symbols of external objects; and the form which we give them is such that the necessary consequences of the images in thought always mirror the images of the necessary consequences in nature of the things pictured”. From these perspectives, physical theories may seem to reflect ontology. However, their core ‘images’ turn out to be epistemic constructions.

In the subsequent discussion, our focus will be on the construction of space-time frames, not in a Newtonian or Kantian sense, portrayed as premeditated ‘as they are’ and providing a sort of theater and arena in which (quantum) events take place, but rather in a Leibnizian sense, constructing them as they can be by the available operational means [10]. As stated by Leibniz [11] (p. 14), “space [[is]] something purely relative, as time is—[[space is]] an order of coexistences, as time is an order of successions”.

Zooming in on the program of ‘it-from-click’ (re)construction of space-time from elementary quantum events, the roadmap is quite straightforward: as quanta are formalized by Hilbert space entities, such an endeavor must somehow ‘translate’ arbitrary dimensional Hilbert spaces into four-dimensional configuration space equipped with space-time frames.

## 2. Conventions and the Necessity of Parameter Independence and, Thus, Choice

We need to be particularly aware of the conventions involved in constructing space-time frames. One such convention is the frame-independent determination of the velocity of light [12,13] in the International System of Units (SI), which means that light cones remain unchanged. Alongside the assumption of bijective mappings of space-time point labels in distinct coordinate frames, this convention, preserving the quadratic distance (Minkowski metric) of zero, leads to affine Lorentzian transformations [14,15].

These conventions formally imply and define the Lorentz transformations of the theory of special relativity. They are inspired by physics, but lack inherent physical content themselves. Their physical significance arises from the preservation of the form invariance of equations of motion, such as Maxwell’s equations, under Lorentz transformations that include (the conventionally defined [12,13] constant and frame-independent) velocity of light.

With regard to synchronization within inertial frames, it is essential to keep in mind that quantum measurements essentially amount to ‘(ir)reversible’ [16,17,18] clicks in some detectors. As long as those detections are statistically independent, we can synchronize time at different locations using radar (‘round-trip’, ‘two-way’) coordinates obtained by sending a (light-in-vacuum) signal back and forth between the respective locations, a procedure known as Poincaré–Einstein synchronization [19,20,21,22,23,24]. As pointed out by Poincaré in 1900 [19] (p. 272) (see also Poincaré’s 1904 paper [20] (p. 311)), suppose that two embedded observers *A* and *B* are positioned at different points of a moving frame, and are unaware of their shared motion, and synchronize their clocks using light signals. These observers believe, or rather assume or define, that the signals travel at the same speed in both directions. They conduct observations involving signals crossing from *A* to *B* and then, vice versa, from *B* to *A*. Their synchronized ‘local’, intrinsic, time can be, according to Einstein [21] (p. 894), defined by (similar) clocks that have been adjusted such that, for the light emission and return times $t_{A}$ and $t_{A}^{\prime }$ at *A*, and the reception and emission time $t_{B}$ at *B*, $t_{B}-t_{A}=t_{A}^{\prime }-t_{B}$. This type of synchronization, if performed with light rays in vacuum, is consistent with the International System of Units (SI) standards.

A formal expression of the statistical independence of two events, outcomes, or observables, *L* and *R*, is the fact that their joint state $\Psi_{LR}$ can be written as the product of their individual states $\Psi_{L}$ and $\Psi_{R}$; that is, $\Psi_{LR}=\Psi_{L}\Psi_{R}$. These states are then nonentangled and separable with respect to observables *L* and *R*.

However, what about entangled states? In this case, independence cannot be assumed as, by definition, the joint state is not a product of the constituent states. Quantum entangled states are encoded relationally [6,25,26]. Since the product rule does not hold for quantum entangled states, we cannot assume that the respective individual outcomes are guaranteed to be mutually separate or mutually distinct in these observables.

## 3. Inseparability and the Lack of Mutual, Relational Choice

The forthcoming argument will contend that entangled quantum states do not appear to provide the means for such spatial order of coexistences, nor for any order of successions. Entangled states lack distinctness between their constituents. A formal expression of such quantum relational encoding is the outcome dependence of two respective events, outcomes, or observations *L* and *R* belonging to the registrations of those entangled particle pairs.

However, outcomes on either side *L* or *R* maintain their statistical parameter independence, which means that any parameter measured at *L* does not affect the outcome or any other operationally accessible observable at *R*, and vice versa. In Shimony’s terminology [27,28], “an experimenter at *R*, for example, cannot affect the statistics of outcomes at *L* by selective measurements”. This can be ensured by the indefiniteness of the respective outcomes, which appear irreducibly random [29] with respect to a range of physical operational means deployable by an intrinsic observer.

State factorization guarantees a specific feature that is crucial for radar coordinates: choice. Simultaneity conventions require the capacity to independently select space-time labels for both types of measurements (parameter independence) and their outcomes, regardless of what is being measured and recorded elsewhere. Outcome independence, along with the resulting temporal and spatial distinctiveness, is essential for establishing any internally operational space-time scale.

Without the freedom to make choices regarding spatiotemporal labeling, the concept of clocks and the measurement of space and time they provide becomes unattainable. Indeed, distinct labels require a distinction among entities to be labeled. However, for quantum entangled states that have traded individuality for relationality, there is no distinction concerning the respective observables.

Suppose, for the sake of demonstration, an isolated mini-universe composed of entangled states, such as the singlet Bell state $|\Psi_{12}^{-}\rangle$ from the Bell basis
(1)$$\begin{array}{c} |\Psi_{12}^{\pm }\rangle =\frac{1}{2}\left(|0_{1}1_{2}\rangle \pm |1_{1}0_{2}\rangle \right),\;|\Phi_{12}^{\pm }\rangle =\frac{1}{2}\left(|0_{1}0_{2}\rangle \pm |1_{1}1_{2}\rangle \right). \end{array}$$

The first and second (from left to right) entries refer to the first and second constituents, respectively. Typically, these constituents are understood to be spatially separated, preferably under strict Einstein locality conditions [30]. For example, Einstein, Podolsky, and Rosen (EPR) employed such spatially separated configurations to argue against the ‘completeness’ of quantum mechanics [31,32].

However, we do not wish to confine ourselves to space-like entanglement. We also aim to encompass time-like entanglement. This type of entanglement can—in the customary space-time frames that we assume to be ad hoc creations of certain nonentangled elements, such as light rays of classical optics, in the standard Poincaré–Einstein protocols mentioned earlier—be generated through processes such as delayed-choice entanglement swapping. Formally, achieving this involves reordering the product $|\Psi_{12}^{-}\Psi_{34}^{-}\rangle$, expressed in terms of the four individual product states $|\Psi_{14}^{+}\Psi_{23}^{+}\rangle$, $|\Psi_{14}^{-}\Psi_{23}^{-}\rangle$, $|\Phi_{14}^{+}\Phi_{23}^{+}\rangle$, and $|\Phi_{14}^{-}\Phi_{23}^{-}\rangle$ of the Bell bases of the ‘outer’ (14) and ‘inner’ (23) particles [33,34,35,36]. Bell state measurements of the latter, ‘inner’ particles yield a rescrambling of the ‘outer’ correlations. Hence, postselecting the ‘inner’ pair (23) results in the desired ‘outer’ Bell states (14), respectively. In more detail, in the Bell basis (Equation 1),
(2)$$\begin{array}{cc} |\Psi_{12}^{-}\Psi_{34}^{-}\rangle & =\frac{1}{2}\left(|\Psi_{14}^{+}\Psi_{23}^{+}\rangle -|\Psi_{14}^{-}\Psi_{23}^{-}\rangle -|\Phi_{14}^{+}\Phi_{23}^{+}\rangle +|\Phi_{14}^{-}\Phi_{23}^{-}\rangle \right),\\ |\Psi_{12}^{+}\Psi_{34}^{+}\rangle & =\frac{1}{2}\left(|\Psi_{14}^{+}\Psi_{23}^{+}\rangle -|\Psi_{14}^{-}\Psi_{23}^{-}\rangle +|\Phi_{14}^{+}\Phi_{23}^{+}\rangle -|\Phi_{14}^{-}\Phi_{23}^{-}\rangle \right),\\ |\Phi_{12}^{-}\Phi_{34}^{-}\rangle & =\frac{1}{2}\left(-|\Psi_{14}^{+}\Psi_{23}^{+}\rangle -|\Psi_{14}^{-}\Psi_{23}^{-}\rangle +|\Phi_{14}^{+}\Phi_{23}^{+}\rangle +|\Phi_{14}^{-}\Phi_{23}^{-}\rangle \right),\\ |\Phi_{12}^{+}\Phi_{34}^{+}\rangle & =\frac{1}{2}\left(|\Psi_{14}^{+}\Psi_{23}^{+}\rangle +|\Psi_{14}^{-}\Psi_{23}^{-}\rangle +|\Phi_{14}^{+}\Phi_{23}^{+}\rangle +|\Phi_{14}^{-}\Phi_{23}^{-}\rangle \right). \end{array}$$

The first of these four equations undergoes careful analysis in References [33,34,35], while the remaining three represent generalizations of this analysis. In the ‘magic’ Bell basis where $|\Psi^{-}\rangle$ and $|\Phi^{+}\rangle$ are multiplied by the imaginary unit *i* [35,37], the relative phases change accordingly.

Delay lines serve as essential components for temporal entanglement. These delay lines could, in principle, also lead to mixed temporal-spatial quantum correlations, where for instance, pairs (12) are spatially entangled while pairs (34) are temporally entangled, resulting in an ‘outer’ pair (14) that is both spatially and temporally entangled. As a consequence, we may consider the particle labels 1,…,4, which have been written as subscripts, to stand for generic spacetime coordinates; that is,
(3)$$\begin{array}{c} 1\equiv \left(x_{1}^{1},x_{1}^{2},x_{1}^{3},x_{1}^{4}=t_{1}\right),\\ 2\equiv \left(x_{2}^{1},x_{2}^{2},x_{2}^{3},x_{2}^{4}=t_{2}\right),\\ 3\equiv \left(x_{3}^{1},x_{3}^{2},x_{3}^{3},x_{3}^{4}=t_{3}\right),\\ 4\equiv \left(x_{4}^{1},x_{4}^{2},x_{4}^{3},x_{4}^{4}=t_{4}\right). \end{array}$$

Equation (Equation 3) is not an ‘equation’ in the strict sense but represents equivalences, as indicated by the equivalence signs. The operationalization of the space-time coordinates referred to in Equation (Equation 3) by radar coordinates, using quasi-classical protocols for quantized systems, is a nontrivial task. However, within the constraints of preparation and measurement, it constitutes a standard procedure already mentioned by Poincaré and Einstein.

We note that temporally entangled shares (as well as mixed temporal-spatial ones) could lead to standard violations of Bell–Boole-type inequalities—for instance, at a single point in space but at different times. The derivation seems to be straightforward: all that is required is a respective Hull computation of the classical correlation polytope [38,39], yielding inequalities that represent the edges of the classical polytope, followed by the evaluation of the (maximal) quantum violation thereof [40,41]. One of the reasons for the seamless transfer of spatial and temporal variables is their interoperability and their realization using delay lines, when necessary.

While considering the question of whether and how such entangled shares could lead to space-time scales, and ultimately frames, or disallows their operational creation, we make three observations: First, the two ‘constituents’ of the relationally entangled share reveal themselves, if compelled into individual events, through two random outcomes that are mutually dependent due to quantum correlations in the form of the quantum cosine expectation laws. These single individual outcomes are expected to be independent of the experiments or parameters applied on the respective ‘other side’ or at the ‘other time’.

Second, these correlations surpass the classical linear correlations [42] for almost all relative measurement directions (except for the collinear and orthogonal directions). However, since these correlations are only dependent on (relative) outcomes and not on parameters, this does not lead to inconsistencies with classical space-time scales generated by the conventional classical Poincaré–Einstein synchronization convention. Indeed, even ‘stronger-than-quantum’ correlations, such as a Heaviside correlation function [43,44] would, under these conditions, not result in violations of causality through faster-than-light signaling.

Third, since individual outcomes cannot be controlled, any synchronization convention and protocol that depends on controlled outcomes cannot be carried out with entangled shares, as there is no means of transmitting (arrival and departure) information ‘across those shares’. Due to parameter independence, any space-time labeling using those outcomes is arbitrary. For instance, ‘synchronizing’ distant clocks (not with light ray exchange, but) by the respective correlated outcomes of entangled particles, such as from spin state or polarization measurements, results in correlated but random temporal scales. These scales cannot be brought into any concordance with ‘local’ time scales generated by the conventional classical Poincaré–Einstein synchronization convention mentioned earlier.

Signaling from one space-time point to another assumes choice, yet again, the form of relational value definiteness that comes at the expense of individual value definiteness, originating from the unitarity of quantum evolution, between two or more constituents of a quantum entangled share prevents signaling across its constituents. Therefore, in the hypothetical scenario of a universe composed of entangled particles, Poincaré–Einstein synchronization may require classical means that are unavailable for entangled particles.

## 4. Orthogonality of Configuration Space from Hilbert Space

Although entanglement does not provide a means for scale synchronization, it can be utilized for synchronizing directions, as well as orthogonality among different frames.

Suppose that all observers agree to ‘measure the same type of observable’, such as spin or linear polarization. It is important to note that, at this stage, we have not yet established a spatial frame. Therefore, for example, an observable like the ‘direction of spin’ (or, for photons, linear polarization) is initially undefined. It must be defined in terms of quantum mechanical entities, such as the state (Equation 1), and observables. Ultimately, this process involves the interpretation of clicks in a detector.

Directional synchronization of spatiotemporal frames can be established, for instance, through the state (Equation 1) by employing successive measurements of particles in that state. In this manner, the directions can be synchronized by maximizing correlations.

Three- and four-dimensionality can also be established by exploiting correlations: (mutual) spatiotemporal orthogonality can be established by (mutually) minimizing the absolute value of these correlations. In this manner, Hilbert space entities are indirectly translated into the orthogonality structure of the configuration space.

## 5. Controllable Nonlocality and Parameter Dependence of Outcomes Due to Nonlinearity of Quantum Field Theory?

We might hope that the addition of nonlinearity via interactions or statistical effects—for example, higher-order perturbation expansions—might help overcome the parameter independence of outcomes in an EPR-type setup. However, as of now, there is no indication of any violation of Einstein locality in field theory [45,46,47,48].

In my earlier publications [49], I have speculated that if one constituent of an EPR pair were to enter a region of high or low density of a particular particle type—for instance, ‘boxes of particles in state |0〉’—then stimulated emission might encourage the corresponding state of the constituent ‘to materialize’ with a higher or lower probability. This, in turn, could be a scenario for the parameter dependence of outcomes, even under strict Einstein locality conditions.

## 6. Summary and Afterthoughts

As argued earlier, there is no independent choice among the individual outcomes of entangled particles: an observer at the ‘one constituent end’ of an entangled share has no ability to select or establish a specific time as a pointer reading.

Nevertheless, it is important to note that not all observables of a collection of particles may be entangled; some could be factorizable. In this case, the latter type of observables may still be applicable for the creation of relativistic space-time frames, unlike the entangled ones.

These considerations are not directly related to the ‘problem of (lapse of) time’ that has led to the notion of a fictitious stationary ‘external’ versus an ‘intrinsic’ time [50,51,52] by equating it with the measurement problem in quantum mechanics.

The adage that “If … two spacetime regions are spacelike separated, then the operators should commute” [8] implicitly supposes two assumptions:(i)First, Einstein’s separation criterion (German ‘Trennungsprinzip’ [53] (pp. 537–539)), which states that relativity theory, and in particular its causal structure determined by light cones, applies to observables formalized as operators.Recall that Einstein, in a letter to Schrödinger [32,53], emphasized (wrongly in my interpretation of the argument) that following a collision that entangles the constituents *L* and *R*, the compound state could be thought of as comprising the actual state of *L* and the actual state of *R*. Einstein argues that those states should be considered unrelated—in particular, there is no relationality. Therefore, the real state of *L* (due to possible spacelike separation) cannot be influenced by the type of measurement conducted on *R*.Our approach diverges from Einstein, insofar as we deny the existence of a preexisting Newtonian space-time theater, even in the modified version proposed by Poincaré and Einstein. Therefore, we cannot depend on a preexisting space-time structure for operators to commute.(ii)Second, it assumes that states are distinct from operators, even though pure states can be reinterpreted as the formalization of observables; specifically, as the assertion that the system is in the respective state.

Since Poincaré–Einstein synchronization via radar coordinates requires a choice and thus parameter dependence, the utilization of entangled states becomes impossible. Hence, we are restricted to separable states. The separability and value definiteness of components within a physical system ultimately reduces to the measurement problem in quantum mechanics. This measurement problem, which involves understanding how an entangled system experiences ‘individualization’ under strictly unitary transformations, with associated value definite information on individual components of the system, remains notoriously unresolved.

We must acknowledge that, at least for now, in the case of relationally encoded entangled quantum states, there is no spatiotemporal resolution. However, due to parameter independence, this type of ‘nonlocality’ cannot be exploited for signaling or radar coordination. Without individuation and measurement, there can be no operational significance assigned to space-time. From this perspective, quantum coordinatization reduces to quantum measurements which, at least in the author’s view, remains unresolved, although it is taken for granted for all practical purposes (FAPP) [54].

A final caveat seems to be in order: The matters and issues discussed in the article could not be fully resolved. However, attempts towards their resolution in terms of entangled systems have been made. One legitimate interpretation is that entangled states cannot be used to construct space-time frames via the Poincaré–Einstein synchronization procedure, resulting in radar coordinates. This might be resolved by adding the particular context of coordinatization and acknowledging means relativity. Thereby, a framework for ‘relativizing relativity’ has been discussed.

## Data Availability

No new data were created or analyzed in this study. Data sharing is not applicable to this article.

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
