# Peer review of "(Re)Construction of Quantum Space-Time: Transcribing Hilbert into Configuration Space"

_entropy, 2024, doi:10.3390/e26030267_

Round 1

Reviewer 1 Report

Comments and Suggestions for Authors

This article discusses an interesting topic, namely, the possible emergence of an effectively classical space-time background from a more fundamental quantum description, expressed in terms of states in a Hilbert space.

Unfortunately, it contains various flaws and shortcomings, which, I believe, make it unsuitable for publication in its present form. My main criticism of this paper is that it does not contain any new mathematical results, or, in fact, any non-textbook mathematical results at all. It contains only three equations. Equation (1) is simply the definition of the Bell basis for a two-fermion system and Eq. (2) gives its extension to four fermions. These are textbook results, while Equation (3) is not really an equation at all, as it has no well-defined mathematical meaning.

The author seems to imply, in Eq. (3) and the surrounding text, that each particle in a four-fermion system can be identified with an abstract space-time point, (x0,x1,x2,x3). But what, exactly, doe this mean? There is, here, no concrete ‘transcription’ from the states of the Hilbert space to the space-time submanifold of the classical configuration space, as implied by the title of the article. There is no well-defined map from one to the other.

The remaining discussion in the text raises some interesting points and issues, but, without a concrete mathematical result at its core, adds little to the existing literature on the emergence of space-time from quantum states. The author’s main point is that entangled states cannot be used to construct space-time frames via the Poincare-Einstein synchronisation procedure. This is undoubtedly true, and worth pointing out, but what then? The paper’s title suggests that the author proposes an alternative construction, or some “quantum generalisation” of the notion of a classical space-time reference frame, but, in fact, he does not attempt this.

Though it is permissible for special issues to admit more speculative articles, and for the usual criteria for the publication of a research article to be relaxed to a certain degree, I do not believe that this article is, ultimately, substantive enough to warrant publication. Regretfully, I cannot recommend it for publication in Entropy.

Author Response

"This article discusses an interesting topic, namely, the possible emergence of an effectively classical space-time background from a more fundamental quantum description, expressed in terms of states in a Hilbert space."

I fully agree with the Referee that the topic of the article is perztinent and well worth studying.

"Unfortunately, it contains various flaws and shortcomings, which, I believe, make it unsuitable for publication in its present form. My main criticism of this paper is that it does not contain any new mathematical results, or, in fact, any non-textbook mathematical results at all. It contains only three equations. Equation (1) is simply the definition of the Bell basis for a two-fermion system and Eq. (2) gives its extension to four fermions. These are textbook results, while Equation (3) is not really an equation at all, as it has no well-defined mathematical meaning."

I agree with the Referee that "Equation (1) is simply the definition of the Bell basis for a two-fermion"; but I need this definition for further discussion and the numbering for reference to it.

I respectfully disagree with the Referee that "Eq. (2) gives its extension to four fermions system". Besides the fact that an extension to four fermions system would require 2^4=16 equations and not four, as displayed in Equation (2), those equations have very specific meanings: the first one of the four equations is subject to a careful analysis in References 31-33, and the remaining three are generalizations of this analysis.

Therefore, in order to avoid such misunderstandings, I have now added the following sentence after Equation (2): 'The first of these four equations undergoes careful analysis in References \cite{Zuk-1993-entanglementswapping Megidish_2013,peres-DelayedChoiceEntanglementSwapping}, while the remaining three represent generalizations of this analysis.'

I also respectfully disagree with the Referee on his evaluation "Equation (3) is not really an equation at all, as it has no well-defined mathematical meaning." Indeed, Equation (3) is no "equation" in the strict sense, but represent equivalences, as indicated by the equivalence signs.

Therefore, in order to avoid such misunderstandings, I have now added the following sentence after Equation (3): 'Equation~(\ref{2023-st-stlabels}) is not an `equation' in the strict sense but represents equivalences, as indicated by the equivalence signs.'

"The author seems to imply, in Eq. (3) and the surrounding text, that each particle in a four-fermion system can be identified with an abstract space-time point, (x0,x1,x2,x3). But what, exactly, doe this mean? There is, here, no concrete ‘transcription’ from the states of the Hilbert space to the space-time submanifold of the classical configuration space, as implied by the title of the article. There is no well-defined map from one to the other."

I am referring to operational space-time coordinates displayed in Equation (3), not to particles. I agree that the operationalization of such points by radar coordinates, using quasi-classical protocols for quantized systems, is a nontrivial task. However, it is a standard procedure already mentioned by Poincaré and Einstein.

Therefore, in order to clarify the situation I have added the following sentence: 'The operationalization of the space-time coordinates referred to in Equation~(\ref{2023-st-stlabels}) by radar coordinates, using quasi-classical protocols for quantized systems, is a nontrivial task. However, within the constraints of measurement, it constitutes a standard procedure already mentioned by Poincaré and Einstein.'

"The remaining discussion in the text raises some interesting points and issues, but, without a concrete mathematical result at its core, adds little to the existing literature on the emergence of space-time from quantum states. The author’s main point is that entangled states cannot be used to construct space-time frames via the Poincare-Einstein synchronisation procedure. This is undoubtedly true, and worth pointing out, but what then? The paper’s title suggests that the author proposes an alternative construction, or some “quantum generalisation” of the notion of a classical space-time reference frame, but, in fact, he does not attempt this."

I fully agree with the Referee about the importance of the issues raised. I also agree that the discussed situation is not fully resolved. However, attempts towards its resolution in terms of entangled systems are being made.

Therefore, I have added the following sentence at the end of the article: 'A final caveat seems to be in order: The matters and issues discussed in the article could not be fully resolved. However, attempts towards their resolution in terms of entangled systems have been made. One legitimate interpretation is that entangled states cannot be used to construct space-time frames via the Poincaré-Einstein synchronization procedure, resulting in radar coordinates. This might be resolved by adding the particular context of coordinatization and acknowledging its means in relativity. Thereby, a framework for 'relativizing relativity' has been discussed."

"Though it is permissible for special issues to admit more speculative articles, and for the usual criteria for the publication of a research article to be relaxed to a certain degree, I do not believe that this article is, ultimately, substantive enough to warrant publication. Regretfully, I cannot recommend it for publication in Entropy."

I hope that, with the additions and explanations mentioned earlier, the manuscript can be published.

I acknowledge, with all due respect, that I may disagree with the Referee regarding the novelty of the discussion and the findings. 

Reviewer 2 Report

Comments and Suggestions for Authors

This manuscript can be regarded as the serial efforts of Refs 1 and 2, which explore the implication of quantum entanglement in constructing space-time frame. Therein, Poincaré–Einstein synchronization is analyzed in terms of spatial and temporal entanglement. Indeed, it deserves its publication in Entropy.

To make the manuscript more friendly to readers, figures can be useful to explain the connection between Poincaré–Einstein synchronization and spatial/temporal entanglement.

Author Response

"This manuscript can be regarded as the serial efforts of Refs 1 and 2, which explore the implication of quantum entanglement in constructing space-time frame. Therein, Poincaré–Einstein synchronization is analyzed in terms of spatial and temporal entanglement. Indeed, it deserves its publication in Entropy."

I kindly thank the Referee for this evaluation.

"To make the manuscript more friendly to readers, figures can be useful to explain the connection between Poincaré–Einstein synchronization and spatial/temporal entanglement."

I respectfully and kindly note that, in this particular context of synchronization by radar coordinates, figures might not, for a considerable portion of the audience, contribute to a further understanding of the discussion and the argument.          

Reviewer 3 Report

Comments and Suggestions for Authors

This paper addresses the question of how spacetime structure can emerge out of the formalism of quantum mechanics. Quantum mechanics operates naturally within Hilbert spaces, of generally large dimension, and how observers of phenomena in such spaces apprehend the usual spacetime as an “epiphenomenon” is the main question studied in this paper. Among the measurements made at the microscopic level out of which spacetime is to be constructed are ones on entangled states, and these pose a special problem. The author illustrates these difficulties by looking at four-particle states that are direct products of Bell states between two pairs and reexpressing in terms of Bell states of particles that were initially uncoupled. His conclusion, after a detailed discussion, is captured in the statement “We must acknowledge that, at least for now, in the case of relationally encoded entangled quantum states, there is no spatiotemporal resolution.”

 The author’s discussion of the challenges posed by this approach is bound to be of interest to others who think about the problem, and for that reason I would recommend publication of this paper.

Author Response

"This paper addresses the question of how spacetime structure can emerge out of the formalism of quantum mechanics. Quantum mechanics operates naturally within Hilbert spaces, of generally large dimension, and how observers of phenomena in such spaces apprehend the usual spacetime as an “epiphenomenon” is the main question studied in this paper. Among the measurements made at the microscopic level out of which spacetime is to be constructed are ones on entangled states, and these pose a special problem. The author illustrates these difficulties by looking at four-particle states that are direct products of Bell states between two pairs and reexpressing in terms of Bell states of particles that were initially uncoupled. His conclusion, after a detailed discussion, is captured in the statement “We must acknowledge that, at least for now, in the case of relationally encoded entangled quantum states, there is no spatiotemporal resolution.” "

The author’s discussion of the challenges posed by this approach is bound to be of interest to others who think about the problem, and for that reason I would recommend publication of this paper.

I kindly thank the Referee for this evaluation. I also refer to the caveat added at the end of the manuscript, which may align with some of the thoughts of the Referee.

Round 2

Reviewer 1 Report

Comments and Suggestions for Authors

I have read the author's reply, and amendments to the text, carefully, but I still struggle to understand how abstract space-time coordinates can be made `equivalent' to particles, in any sense, and particularly when one restricts measurements on the latter only to spin. In his reply, the author refers to these as "operational space-time" coordinates and states that "'The operationalization of the space-time coordinates referred to in Equation (3) . . . constitutes a standard procedure already mentioned by Poincaré and Einstein". I am not aware of any work by either Einstein, or Poincaré, which associates abstract space-time points with the spin-states of material particles. If such a work exists, the author should clearly reference it at the point in the text where these claims are made.

For this reason, I still have quite strong concerns about scientific content of this article. Nonetheless, as a conference proceedings article, its main purpose is give an accurate record of what material the author did, in fact, present at the conference in question. Such presentations can, legitimately, be more speculative and less rigorous than a standard research article. Therefore, overall, I am able still to recommend publication, even with the reservations noted above. However, before final publication, I must insist that the work by Einstein, or Poincaré, referred to, but not clearly referenced, in the new draft, is explicitly included in the Bibliography. This will allow readers who are not familiar with it, like myself, to check it for themselves.

Comments on the Quality of English Language

There are no significant problems regarding the English language of the text. I recommend only a final spelling and grammar check before publication.

Author Response

I agree with the Referee that there is no "work by either Einstein, or Poincaré, which associates abstract space-time points with the spin-states of material particles."

Therefore, I have enlarged the discussion on possible synchronization conventions using entangles shares a bit as follows:

"Due to outcome dependence yet parameter independence, any space-time labeling using those outcomes is arbitrary.
For instance, `synchronizing' distant clocks (not with light ray exchange but) by the respective correlated outcomes of entangled particles
results in correlated but random temporal scales, which cannot be brought into any concordance
with `local' time scales generated by the conventional classical Poincar\'e-Einstein synchronization convention.

Signaling from one space-time point to another assumes choice,
yet again, the form of relational value definiteness that comes at the expense of individual value definiteness,
originating from the unitarity of quantum evolution,
between two or more constituents of a quantum entangled share
prevents signaling across its constituents.
Therefore, in the hypothetical scenario of a universe composed of entangled particles,
Poincar\'e-Einstein synchronization may require classical means that are unavailable for entangled particles."

I have also added

* two references;

* a more detailed explanation of Poincare-Einstein synchronization, quoting those new references, and Einstein's 1905 paper:

"As pointed out by Poincar\'e in 1900~\cite[p.~272]{Poincare1900} (see also Poincar\'e's 1904 paper~\cite[p.~311]{Poincare1904}),
suppose that two embedded observers $A$ and $B$ are positioned at different points of a moving frame, and are unaware of their shared motion,
and synchronize their clocks using light signals.
These observers believe, or rather assume or define, that the signals travel at the same speed in both directions.
They conduct observations involving signals crossing from $A$ to $B$ and then, vice versa, from $B$ to $A$.
Their synchronized `local', intrinsic, time can be, according to Einstein~\cite[p.~894]{ein-05},
defined by (similar) clocks that have been adjusted such that, for the light emission and return times $t_A$ and $t_A'$ at $A$,
and the reception and emission time $t_B$ at $B$, $t_B - t_A = t_A' - t_B$. ";

* as well as eliminated the bracket "(in a zig-zag manner)".  

Round 3

Reviewer 1 Report

Comments and Suggestions for Authors

As a record of what the author talked about at a conference, this article is acceptable in its present form. In my view, it still falls a long way short of the rigour required from a standard research article. Nevertheless, as a record of the conference proceedings, it is a legitimate and acceptable contribution to the existing literature.